# The Interplay between Colon Cancer Cells and Tumour-Associated Stromal Cells Impacts the Biological Clock and Enhances Malignant Phenotypes

**DOI:** 10.3390/cancers11070988

**Published:** 2019-07-15

**Authors:** Luise Fuhr, Mónica Abreu, Annalucia Carbone, Rukeia El-Athman, Fabrizio Bianchi, Mikko O. Laukkanen, Gianluigi Mazzoccoli, Angela Relógio

**Affiliations:** 1Institute for Theoretical Biology (ITB), Charité—Universitätsmedizin Berlin, Corporate Member of Freie Universität Berlin, Humboldt-Universität zu Berlin and Berlin Institute of Health, 10117 Berlin, Germany; 2Molekulares Krebsforschungszentrum (MKFZ), Charité—Universitätsmedizin Berlin, Corporate Member of Freie Universität Berlin, Humboldt-Universität zu Berlin and Berlin Institute of Health, 10117 Berlin, Germany; 3Division of Internal Medicine and Chronobiology Unit, Fondazione IRCCS (Istituto di Ricerca a Carattere Clinico e Scientifico) Casa Sollievo della Sofferenza, 71013 San Giovanni Rotondo (FG), Italy; 4Unit of Oncology Biomarkers, Fondazione IRCCS (Istituto di Ricerca a Carattere Clinico e Scientifico) Casa Sollievo della Sofferenza, 71013 San Giovanni Rotondo (FG), Italy; 5Research laboratory, Pineta Grande, 80131 Caserta, Italy

**Keywords:** cancer, colorectal, tumour stroma, biological clock

## Abstract

Cancer cells interrelate with the bordering host microenvironment that encompasses the extracellular matrix and a nontumour cellular component comprising fibroblasts and immune-competent cells. The tumour microenvironment modulates cancer onset and progression, but the molecular factors managing this interaction are not fully understood. Malignant transformation of a benign tumour is among the first crucial events in colorectal carcinogenesis. The role of tumour stroma fibroblasts is well-described in cancer, but less well-characterized in benign tumours. In the current work we utilized fibroblasts isolated from tubulovillous adenoma, which has high risk for malignant transformation, to study the interaction between benign tumour stroma and the circadian clock machinery. We explored the role of the biological clock in this interplay taking advantage of an experimental model, represented by the co-culture of colon cancer cells with normal fibroblasts or tumour-associated fibroblasts, isolated from human colorectal tumour specimens. When co-cultured with tumour-associated fibroblasts, colon cancer cells showed alterations in their circadian and metabolic parameters, with decreased apoptosis, increased colon cancer cell viability, and increased resistance to chemotherapeutic agents. In conclusion, the interactions among colon cancer cells and tumour-associated fibroblasts affect the molecular clockwork and seem to aggravate malignant cell phenotypes, suggesting a detrimental effect of this interplay on cancer dynamics.

## 1. Introduction

Colorectal cancer (CRC) ranks the fourth leading cause of cancer deaths and the third most commonly diagnosed malignant neoplastic disease [1,2,3]. CRC develops gradually and initiates from premalignant polyps such as tubular adenomas, tubulovillous adenomas, sessile serrated adenomas, and traditional serrated adenomas [4]. Pre-neoplastic tissues of gastrointestinal cancers contain parenchymal transformed cells, as well as multiple host stromal cells [5]. Mounting evidence indicates that tumour stroma, more than passively hosting and supporting the mutated epithelial cells, impacts and/or even hastens colorectal carcinogenesis. The tumour microenvironment seems to play a crucial role in CRC progression, metastasis, and drug resistance. It consists of an extracellular matrix, tumour-associated fibroblasts (TAFs) able to uphold carcinogenesis and dwindle drug sensitivity, activated immune cells, and tumour vasculature [6]. These components interrelate with the cancer cells by means of a series of secreted molecules supporting paracrine signalling, as well as cell-to-cell contact mechanisms, creating a bi-univocal communication array [7,8,9,10]. In particular, the signals originating from CRC stromal cells take part in triggering specific gene expression signatures in tumour parenchymal cells, which sequentially prompt deranged gene expression profiles in stromal cells [11]. The tumour microenvironment changes during cancer progression in line with unremitting modification of the signals originating from the different components of the microenvironment, provoking altered responses from the adjacent cells. Moreover, colorectal tumorigenesis is supported by alteration of essential cellular processes, such as cell cycle, proliferation and apoptosis, which are hallmarked by periodic fluctuations. Rhythmicity of biological processes is driven at the cellular level by molecular clockworks operated by transcriptional–translational feedback loops (TTFLs) [12,13]. In mammals and humans, these TTFLs are mainly driven by the heterodimer complex formed by the proteins BMAL1/ BMAL2 and CLOCK/NPAS2, which activate the transcription of *PERIOD* (*PER1-3*), *CRYPTOCHROME* (*CRY1-2*), *REV-ERB* (*REV-ERB1*, *REV-ERB2*), and *ROR* (*RORa*, *RORb*, *RORc*) genes and their corresponding proteins that fine-tune the transcriptional activity of BMAL1/2-CLOCK/NPAS2 heterodimers [14,15]. The biological clock drives crucial cellular processes, such as the cell division cycle, metabolism, proliferation, DNA damage response, apoptosis, and autophagy, which when altered are critical in cancer onset and progression [16,17,18,19,20]. Altered expression of *PER* genes was found in non-small lung cancer patients [21], as well as in breast cancer patients. In the latter neoplastic disease *PER1-3* downregulation predicted poorer survival [22]. Cryptochrome proteins regulate cell cycle progression, and their deficiency accelerates cancer cell proliferation [23], and enhances resistance to chemotherapeutic agents [24]. Additionally, *CLOCK* gene upregulation predicts poorer outcome in CRC patients, upholds colon cancer cell proliferation, and reduces apoptosis [25]. BMAL1 is necessary for the p53-dependent stimulation of p21^(Cip1/Waf1)^ [26] and *BMAL1* deficiency hinders p53-dependent cell cycle arrest triggered by DNA damage [27]. BMAL1 hinders the G2/M transition activating *WEE1* kinase expression, with successive inhibition of C*DK1*, and inhibits c-MYC reducing cyclin E expression and cyclin E/Cdk2 activity, hampering the G1/S transition [28,29]. Severe alterations of the biological clock functioning have been reported in the tumour tissue of CRC patients and in colon cancer cells [30,31,32,33,34,35,36,37]. Thus, the aim of our study was to evaluate the putative impact of the interplay between colon cancer cells and stromal cells on the function of the biological clock. To investigate this hypothesis, we accessed the expression patterns of the core-clock genes, as well as the metabolic and phenotypic characteristics of colon cancer cells upon interaction with stromal cells. We explored this cross-talk using co-cultures of colon cancer cells (HCT116) and a commercial line of normal human intestinal fibroblasts (HIFs), as well as primary normal fibroblasts (NFs) and primary tumour-associated fibroblasts (TAFs) derived from specimens obtained from CRC patients. Our results bring forward an important role of cell-to-cell communication in the regulation of the circadian phenotype, with subsequent alterations in mitochondrial respiration, cellular apoptosis, and cytotoxicity.

## 2. Results

### 2.1. Co-Culture of Cell Lines with Different Clock Phenotypes Affects the Expression of Core-Clock Genes

A tumour is a complex tissue, which is composed of different cell types mainly derived from the neighbouring stroma that constitute the tumour microenvironment. It is still unclear how the individual molecular clocks of different cell types in a complex tissue may influence each other. We therefore carried out co-culture experiments and evaluated the effect on the clock phenotype via bioluminescence measurements. In a first attempt, the colorectal cancer cell line HCT116 and the human normal intestinal fibroblast cell line HIF were co-cultured. Colorectal cancer cell lines display a variety of circadian phenotypes, which range from a completely disrupted clock to less dramatic phenotypes, as shown in previous work [38]. For this study, we aimed at using a cell line with a robust oscillator (well-defined circadian phenotype) to evaluate how such an oscillator may be affected by the microenvironment, including primary tumour cells from patients. Since the HCT116 cell line represents such a robust oscillator, as compared to other colorectal cancer cell lines, it was selected for this study (Appendix A). Subsequently, we measured the clock phenotype in each cell type or in co-cultured cells, but with only one cell type being transduced with a construct containing the *BMAL1* promoter and the luciferase coding sequence (BLH) (i.e., HIF-BLH cells or HCT116-BLH cells). In a second attempt, HCT116 cells were co-cultured with NFs or TAFs. In this case, all cells were measured individually and in co-culture, and we analysed the clock phenotype of HCT116 cells to evaluate whether co-culture with stromal cells changed the oscillation profile. Under control conditions, HCT116 and HIF cells showed significantly different periods (*p* < 0.05) (Figure 1A,D). Although the co-culture of both cell lines did not lead to significant changes in period length or phase, the oscillatory patterns changed in HIF-BLH cells upon co-culture with HCT116 cells (Figure 1D,E). In particular, the oscillations were more robust when cells were measured in co-culture (Figure 1B). This effect could not be observed when HCT116-BLH cells were measured in co-culture with HIF cells (Figure 1C–E).

To further explore the functioning of the circadian clock, we evaluated time-course measurements of mRNA expression levels of a number of core-clock and clock-controlled genes in HCT116 and HIF cells, both individually and in co-culture (Figure 2A,B). Samples were collected within a time interval of 18 h. The first sample was collected 20 h after synchronization and the last sample 38 h after synchronisation. Most genes showed variations in their expression values over time in all three conditions with the exception of *CLOCK* in HIF cells and *TIMELESS* in HCT116 cells alone and HCT116 in co-culture (Figure 2A). For nearly all genes, the expression in HIF cells reached its lowest level at 32 h, while in the HCT116-HIF co-culture, the maximum was predominantly at 38 h (Figure 2A,B). When comparing the expression patterns, the cluster of genes containing *CLOCK*, *CRY1*, *PER2,* and *PER3* showed a strongly different trend of regulation in HCT116-HIF co-culture conditions in respect to HCT116 cells alone (Figure 2B). For most of these genes, this trend in regulation appeared to be inverted (Figure 2A). Contrariwise, the expression pattern of other core-clock genes, such as *BMAL1*, *REV-ERBα*, *PER1,* and *TIMELESS*, were similar in HCT116 cells and in the HCT116-HIF co-culture, but they differed in HIF cells. Interestingly, the expression of *RORA* was severely damped under co-culture conditions (Figure 2B). In addition to the changes observed in the bioluminescence recordings experiments, these results suggest that the interplay between two cell types influences the molecular clockwork at the gene expression level, likely to affect protein expression as well, as observed for the oscillatory phenotype of the protein SIRT1 (Appendix A and Appendix A).

### 2.2. Co-Culture with Primary Fibroblasts Induces Circadian Changes in HCT116 Cells

Next, HCT116 cells were co-cultured with primary NFs or TAFs, and the effect on the circadian phenotype was evaluated. We first determined the oscillatory profile, and the circadian parameters period, and phase in single cell-type assay. The period of NFs and TAFs was significantly longer than the period of HCT116 cells (Figure 3A,B) as observed also for the HIF cells (Figure 1). Furthermore, the phase was significantly different in NFs and TAFs as compared to HCT116 cells (Figure 3C). Interestingly, when HCT116-BLH were co-cultured with primary NFs and TAFs, we observed a significantly longer period in HCT116 cells (Figure 3D,E) similar to the period observed for NFs and TAFs. Lastly, we also observed a phase shift as compared to the normal HCT116 oscillatory profile without co-culture (Figure 3F). 

Additionally, we carried out a new co-culture of the HCT116 cells with primary cells of a second patient (with moderately differentiated colon adenocarcinoma) and also observed that, in this new co-culture scenario, the cells from the cancer patient altered the clock phenotype of the HCT116. In particular, there was a significant phase shift of 4.7 h as a result of the co-culture with NF and of 6.2 h upon co-culture with TAF. Furthermore, the period of the oscillations showed alterations, although not significant (Appendix A).

Given that normal and tumour fibroblasts secrete soluble factors, which regulate a wide spectrum of biological processes including proliferation, migration, differentiation, and apoptosis [5], we wondered whether cytokines could play role in this intercellular communication. Of note, TNF-α (i.e., a multifunctional pro-inflammatory cytokine) is rhythmically encoded by *TNF* that is a clock-controlled gene, and in turn alters the functioning of the biological clock [39]. We previously showed that TNF-α and circadian gene modulation alters the expression of clock genes and triggers phenotypic changes in lymphoma cells [40]. We therefore hypothesized that TNF-α could mediate some of the effects observed in the oscillatory profile of HCT116 cells upon co-culture with fibroblasts. We evaluated the responses of HIF and HCT116 to increasing doses of TNF-α and found interesting results: while no change was observed in the circadian phenotype of HIF cells, HCT116 cells showed an increase in period length and a phase shift (Figure 4 and Appendix A).

### 2.3. Circadian Changes Induced by Primary Fibroblasts Impact Metabolism, Proliferation, Viability, Cytotoxicity, and Apoptosis in HCT116 Cells

Previous findings revealed that alterations in circadian rhythm and clock-regulated genes may impinge on the metabolism of cancer cells, and that metabolic switching is a hallmark of an increased aggressiveness [20]. Therefore, in a next step we tested whether the co-culture of colon cancer cells with NFs and TAFs could also influence metabolic activity with impact on cell viability, cytotoxicity, and apoptosis. Glycolysis, measured as the extracellular acidification rate (ECAR) reached by a given cell after the addition of saturating amounts of glucose, was comparable in the three examined conditions (HCT116 cells alone, HCT116 cells + NF, and HCT116 cells + TAF) (Figure 5A). The glycolytic capacity, measured as the maximum ECAR reached by a cell following the addition of oligomycin, effectively shutting down oxidative phosphorylation and driving the cell to use glycolysis to its maximum capacity, was decreased in HCT116 cells and NF co-culture (Figure 5B). This condition was also characterized by decreased glycolytic reserves, indicating the capability of a cell to respond to an energetic demand, as well as how close the glycolytic function is to the cell’s theoretical maximum. Non-glycolytic acidification, measuring other sources of extracellular acidification not attributable to glycolysis, was decreased in both co-culture conditions (Figure 5C). 

Co-culture with NFs or TAFs did not induce changes regarding the glycolytic activity in HCT116 cells (Figure 5A–E), whereas the mitochondrial activity was significantly altered (*p* < 0.05, Figure 6). Basal respiration, a measure of oxygen consumption used to meet cellular ATP demand resulting from mitochondrial proton leak and showing baseline cell energetic demand, and maximum respiration, significantly increased after co-culture with NFs in HCT116 cells (Figure 6A,B). This condition was also characterized by increased ATP production measuring the decrease in oxygen consumption rate upon injection of the ATP synthase inhibitor oligomycin (Figure 6C). It represents the portion of basal respiration used to drive ATP production and shows ATP produced by the mitochondria that contributes to meeting the energetic needs of the cell. During co-culture of HCT116 and NF cells, we also observed increased maximal respiration, measuring the maximal oxygen consumption rate attained by adding the uncoupler FCCP, which mimics a physiological “energy demand” by stimulating the respiratory chain to operate at maximum capacity, causing rapid oxidation of substrates (sugars, fats, amino acids) to meet this metabolic challenge. Similarly, in this condition we observed increased spare respiratory capacity, indicating the capability of the cell to respond to an energetic demand, as well as how close the cell is to respiring to its theoretical maximum. The ability of the cell to respond to energetic demands can be an indicator of cell fitness or flexibility, as well as increased non-mitochondrial respiration (oxygen consumption that persists due to a subset of cellular enzymes that continue to consume oxygen after rotenone and antimycin-A addition). This is important for obtaining an accurate measure of mitochondrial respiration. Proton (H^+^) leak, indicating the remaining basal respiration not coupled to ATP production and an indicator of mitochondrial damage or of a mechanism to regulate the mitochondrial ATP production, was greatly increased when HCT116 cells where co-cultured with TAF (Figure 6D). On the other hand, co-culture with TAFs tendentially decreased the mitochondrial activity (Figure 6A–H). Co-culture with NFs induced a more energetic phenotype in HCT116 cells, while the energy phenotype after co-culture with TAFs was only slightly changed (Appendix A). 

Since circadian changes and TAFs are both known to influence cell viability, apoptosis, and resistance to cytotoxic agents of cancer cells, we therefore investigated the functional implications of this co-culture scenario regarding cytotoxicity, viability, and apoptosis. We measured live-cell protease activity using a fluorogenic, cell-permeant peptide substrate; dead-cell protease activity using a cell-impermeant, fluorogenic peptide substrate; and caspase-3/7 activation as a key indicator of apoptosis. The co-culture of HCT116 cells with TAF lead to an increase of cancer cell viability and resistance to cytotoxic agents, whereas apoptosis significantly decreased. Notably, the co-culture with NF also increased resistance to cytotoxic agents and decreased apoptosis (Figure 7). 

Taken together the results attained from our work showed that cell-to-cell communication of primary fibroblasts and HCT116 cells influenced the circadian phenotype, changed mitochondrial respiration, prevented cells from apoptosis, and increased resistance to cytotoxic agents.

## 3. Discussion

Currently, it is common in oncological research to carry out experiments in human cancer cell lines representing only one type of tumour cell. However, a tumour is a complex tissue composed of different cell types mainly derived by the neighbouring stroma, which constitute the tumour microenvironment [32]. Colorectal cancer tissue contains parenchymal tumour cells and host stromal cells. Tumour stroma acts as an essential modulator and/or even a driver of tumorigenesis, in addition to its role as a supportive structure harbouring neoplastic cells. Among various cell types, including endothelial and immune cells, stromal cells also comprise fibroblasts that may transform into TAFs able to support tumorigenesis and impair drug sensitivity. Interactions between these different cell types influence tumourigenesis [32]. In normal colonic mucosa, the stromal population mainly consists of fibroblasts [41]. During tumourigenesis, through different pathways, fibroblasts may differentiate into TAFs, which become the main cellular component of tumour stroma [42]. TAFs secrete growth factors and thereby support tumour growth and migration. They promote survival and proliferation in primary tumours, as well as in metastasis [43]. Given the known importance of the tumour microenvironment and its constituents, TAFs might be particularly relevant as diagnostic and prognostic biomarkers and should be considered as potential targets for anticancer therapy. It is important to notice that stroma cells, as used in our study, are different from stem cells, and even though the role of stem cells in malignancy is an extremely interesting topic, it is out of the focus of the current study. A recent work by Matsunaga and colleagues [44] addresses the relevance of the molecular clock in the microenvironment for cancer development in a mouse model, and their work shows that cells with a properly ticking biological clock that populate the tumor microenvironment could rhythmically and spontaneously regulate clock-silenced breast cancer stem cells through the circadian release of diffusible factors (i.e., WNT signaling components). The authors further demonstrate that an optimized dosing schedule based on circadian dynamics of mouse breast cancer stem cells improves the antitumor effects of the drug [44]. Cancer stem cells (CSCs), also known as tumour-initiating cells (TICs), are cancer cells that possess capabilities of proliferation, differentiation, and self-renewal (i.e., CSCs are more aggressive cancer cells and are different from normal tissue adult stem cells, such as mesenchymal stem cells). Cancer stroma contains mesenchymal stem cells (MSCs) that can function as an origin of myofibroblasts, also known as cancer-associated fibroblasts (CAFs), which then support cancer progression [45,46]. MSCs present in the tumour stroma, and when damaged (e.g., by cellular stress), can form a direct contact with cancer cells and serve as a source of nutrients that then significantly increases cancer cell growth capacity, metastasis ability, and resistance to cancer drugs.

### Cell-to-Cell Communication Impacts on the Circadian Phenotype

Bidirectional interactions among stromal and parenchymal cancer cells operated by secreted signalling molecules turn on specific gene expression signatures and set up network circuits hardwiring the tumour microenvironment [32]. Long-term, real-time bioluminescence recording performed in various colon cancer cell lines showed different profiles with increasing pattern derangements in cell lines with increasingly malignant phenotypes [38]. 

Based on this knowledge, we evaluated the impact of cell-to-cell communication on the circadian phenotype of HCT116 colon cancer cells upon co-culture with normal and tumour fibroblasts. In our study, oscillatory patterns rendered by long-term, real-time bioluminescence recordings suggested different functioning of the biological clock in the examined colon cancer cells (HCT116 cell line) as well as in normal and tumour fibroblasts. Co-culture induced both a phase shift and a longer period in HCT116 cells. Interestingly, the longer period of HCT116 cells after co-culture closely resembled the period length of NFs and TAFs, suggesting that fibroblasts influenced the period length in this cancer cell line into the direction of their own period length. Neoplastic cells and bordering stromal cells in the tumour microenvironment interact either through direct contact or by means of chemokine and cytokine signalling that impacts cancer cell phenotype (survival, apoptosis, response to cytotoxic agents), as well as neoplastic disease outcome (eradication or progression). We challenged our in vitro model with TNF-α, and the results obtained suggest a direct impact on the functioning of the biological clock and presumably on cellular processes regulated by the molecular clockwork. It would be interesting to investigate, in future work, if the NFs or the TAFs effectively secrete TNF-α or other cytokines, which may be involved in cell-to-cell communication and may alter the circadian phenotype of the surrounding cells. In particular, TNF-α treatment induced a longer period and a phase shift of HCT116 cells, which suggested a potential path via which fibroblasts can impact circadian clock functioning in HCT116 cells. The different patterns of bioluminescence oscillations were paralleled by different patterns of expression of core-clock genes (*BMAL1*, *CLOCK*, *CRY1-2*, and *PER1-3*) and clock-controlled genes (*WEE1*, *cMYC*) assessed in time-course gene expression experiments at the mRNA level in the examined cell types when evaluated alone and in co-culture. The results are in line with existing findings that co-culture of fibroblasts and other cells has a profound impact on the expression of circadian genes [6]. The observed effects on the circadian phenotype could be extended to the metabolic activity, cytotoxicity, and apoptosis. The co-culture of HCT116 cells with primary NF and TAF triggered important metabolic changes, improved colon cancer cell viability, and reduced response to chemical agents. While glycolysis was not affected in HCT116 cells upon co-culture, mitochondrial activity was altered. Basal respiration, maximum respiration, and ATP production were significantly increased after co-culture with NFs in HCT116 cells. On the other hand, co-culture with TAFs decreased the mitochondrial activity. These results suggest that the surrounding stroma regulates metabolic activity in cancer cells, and the effects on cell metabolism may differ based on the type of surrounding cells. Furthermore, cytotoxicity and apoptosis were both significantly decreased in HCT116 cells when co-cultured with NFs or TAFs, pointing to a putative role of the surrounding stroma in protecting cancer cells from cell death and promoting cell survival and growth. It is relevant to point out that in this study we discuss the effect of TAFs prior to, or at the moment of, malignant transformation. Neoplastic polyps, including tubular adenomas, villous adenomas, and tubulovillous adenomas, are precursors of colon cancers. In particular, patients with tubulovillous adenomas are at very high risk to acquire malignant lesions. This hints that epithelial cells in these adenomas can frequently undergo malignant transformation. Neither the role of TAFs in this transformation nor the effect of TAFs on epithelial cells that are about to, or have just undergone malignant transformation are known.

The insights gained from the co-culture experiments show that cancer cells are highly affected by surrounding cells not only with respect to their circadian phenotype, but also with respect to their metabolic activity and cell survival. Our results hint to a modification in the regulation of the molecular clockwork functioning triggered by an interplay between neoplastic and tumour-derived stromal cells, which can ultimately modify colon cancer cell phenotypes.

## 4. Materials and Methods

### 4.1. Cell Culture

CaCo2, HCT116, HT29, LIM1215, RKO, SW480, and SW620 cells were maintained in DMEM low-glucose culture medium (Lonza, Basel, Switzerland supplemented with 10% FBS (Life Technologies, Thermo Fisher Scientific, Waltham, MA, USA), 1% penicillin–streptomycin (Life Technologies), 2 mM Ultraglutamine (Lonza), and 1% HEPES (Life Technologies). Normal colon fibroblasts and tubulousvillous adenoma-associated fibroblasts (TAFs) were derived from the same patient. TAFs were isolated from an ulcerated and bleeding neoformation that was diagnosed as tubulovillous adenoma with low-grade dysplasia and reactive hyperplasia in the local lymph nodes. Normal colon counterpart fibroblasts were isolated at a distance of 20 cm from the adenoma. Samples were collected from two patients using the same procedure (the samples from one patient, described above, are referred as NF and TAF, and from the other patient, diagnosed with moderately differentiated colon adenocarcinoma, as NF2 and TAF2). Human normal colon intestinal primary fibroblasts (HIFs) were purchased from Cliniscience (Cliniscience, Nanterre, France). HIF cells and normal and tumour-associated fibroblasts were maintained in MEM-α (Gibco, Thermo Fisher Scientific, Waltham, MA, USA) medium supplemented with Hyclone FBS (GE healthcare, Munich, Germany), 1× antibiotic–antimycotic (Gibco), and 1× MEM nonessential amino acids solution (Gibco). The ethical permission for the study was approved by IRCCS (Istituto di Ricerca a Carattere Clinico e Scientifico) SDN (Comitato Etico per la Sperimentazione Clinica Progetto N:ro 2013-01-02) and Monaldi Hospital ethical committees (Deliberazione del Direttore Generale n:o 1239).

### 4.2. Lentivirus Production

Lentiviral elements containing a *BMAL1*-promoter-driven luciferase (BLH) were generated as briefly described: HEK293T cells were seeded in 175 cm^2^ culture flasks and co-transfected with 12.5 µg packaging plasmid psPAX, 7.5 µg envelope plasmid pMD2G, and 17.5 µg *BMAL1*-promoter (BLH)-luciferase expression plasmid using the CalPhos mammalian transfection kit (Clontech, Mountain View, CA, USA) according to the manufacturer’s instructions. To harvest the lentiviral particles, the supernatant was centrifuged at 4100× *g* for 15 min to remove cell debris and passed through a 45 µm filter. The lentiviral particles were stored at −80 °C.

### 4.3. Transduction with Lentiviral Vectors

For lentiviral transduction, cells (i.e., HCT116, HIF, primary NF, and TAF) were seeded in 6-well plates. On the day of transduction, 1 mL medium and 1 mL of lentiviral particles was added. A total of 8 µg/mL protamine sulfate (Sigma-Aldrich, St. Louis, MO, USA) and 4 µg/mL polybrene (Sigma) was used to enhance transduction efficiency. The next day, the medium was replaced, and selection medium was added one day later (complete growth medium containing 100 µg/mL hygromycin B) (Gibco) to obtain stable transduced cells and incubated at 37 °C with 5% CO_2_ atmosphere.

### 4.4. Cell Culture for Bioluminescence Measurements

Stable-transduced cell populations were selected and maintained in medium containing hygromycin B (100 µg/mL, Gibco). For live-cell bioluminescence recording, cells were maintained in phenol red-free DMEM (Gibco) containing 10% FBS (Life Technologies), 1% penicillin–streptomycin (Life Technologies), and 250 µM D-Luciferin (PJK, Kleinblittersdorf, Germany). HIF cells and normal and tumour-associated fibroblasts were maintained in phenol red-free MEM-α (Gibco) medium supplemented with 10% Hyclone FBS (GE healthcare), 1× antibiotic–antimycotic (Gibco), and 1× MEM nonessential amino acids solution (Gibco). Cell morphology and density were controlled by light microscopy. All cells were incubated at 37 °C in a humidified atmosphere with 5% CO_2_. Cells were synchronised by medium change prior to measurement. *Bmal1*-promoter-(BLH)-reporter activity was measured using a LumiCycle instrument (Actimetrics, Wilmette, IL, USA) for five consecutive days. Bioluminescence data were analysed using the Chronostar software (version 3.0, no commercial software) [47]. 

### 4.5. Co-Culture, Synchronization, and Bioluminescence Measurements of Circadian Rhythms 

One day prior to the start of the co-culture, cells were seeded at the appropriate density, dependent on the cells used. Un-transduced cells were seeded in 35 mm dishes, and transduced cells (*BMAL1*-promoter-(BLH)-reporter) were seeded on 22 mm coverslips. The next day, cells on coverslips were added to the appropriate un-transduced cells. From this timepoint on, all cells were maintained in MEM-α medium. Co-culture was maintained for 48 h. For long-term, real-time bioluminescence recording, cells were synchronized by serum shock with medium containing 50% FBS for 2 h. Next, cells were washed once with 1× PBS (Life technologies), and phenol red-free MEM-α (Gibco) supplemented with 250 µM D-Luciferin (PJK) was added. *BMAL1*-promoter-(BLH)-reporter activity was measured, using a LumiCycle instrument (Actimetrics) for 5 d. Raw bioluminescence time-series data were detrended with a moving average method, the 24 h running average, using the ChronoStar analysis software [47]. 

### 4.6. RNA Extraction and First-Strand cDNA Synthesis

Total RNA was extracted with TRIzol reagent (Invitrogen, Thermo Fisher Scientific, Karlsruhe, Germany) from HCT116 and HIF cells. The amount of total RNA was determined by UV spectrophotometry using the Nano Drop Spectrophotometer (Nanodrop Technology, Thermo Fisher Scientific, Waltham, MA, USA), and RNA integrity was assessed by the Agilent 2100 Bioanalyzer (Agilent Technologies, Santa Clara, CA, USA) after digestion by DNaseI. Next, 1.0 µg of total RNA was reversed transcribed using the High-Capacity cDNA Archive Kit following the manufacturer’s instructions (Applied Biosystems, Thermo Fisher Scientific, Waltham, MA, USA). 

### 4.7. Quantitative Real-Time Reverse Transcription PCR Assay 

To assess the differential expression of the core-clock genes and clock-controlled genes in HCT116 and HIF cells assessed alone and HCT116 cells in co-culture, quantitative real-time PCR (q-PCR) assay was performed by using *BMAL1* (*ARNTL*, QT00011844), *BMAL2* (*ARNTL2*, QT00068250), *CLOCK* (QT00054481), *CRY1* (QT00025067), *CRY2* (QT00094920), *CSNK1E* (QT00999152), *REV-ERBα* (*NR1D1*, QT00000413), *PER1* (QT00069265), *PER2* (QT00097713), *PER3* (QT00011207), *RORA* (QT00072380), *SIRT1* (QT00051261), *TIMELESS* (QT00019789), *TIPIN* (QT00054334), *WEE* (QT00038199), and *c-MYC* (QT00035406) Human QuantiTec Primers Assay (SYBR Green QuantiTect Primers Assay; QIAGEN, Hilden, Germany). All qPCRs were performed in a 25 µL final volume, with three replicates per sample, by using QuantiFast SYBR Green PCR kit (QIAGEN) and ran in an ABI PRISM^®^ 7700 Sequence Detection System (Applied Biosystems, Thermo Fisher Scientific, Waltham, MA, USA). The data were analysed using the default and variable parameters available in the SDS software package (version 1.9.1; Applied Biosystems). The housekeeping gene *GAPDH* was used as control to normalize the target gene expression levels. The log_2_ fold change in expression compared to the expression at time point 20 h was calculated with the 2^−ΔΔCT^ method. Two biological replicates were each assayed in triplicate, and results were expressed as mean ± standard deviation (SD). Hierarchical clustering analysis was performed on qPCR data normalized as previously described [48]. The constant reference (*K*) for data normalization was obtained by averaging the Cq of GAPDH across all samples (*K* = 14.67). Clustering analysis was performed on Cq-normalized and median-centred settings, using uncentered correlation and centroid linkage (Cluster 3.0 for Mac OS X; http://bonsai.hgc.jp/~mdehoon/software/cluster/software.htm), and visualized by Java TreeView (http://jtreeview.sourceforge.net).

### 4.8. Western Blotting

For protein isolation, cells were gently detached from the dish, sedimented by low-speed centrifugation, and resuspended in lysis buffer. Aliquots containing 30 µg of proteins from each cell lysate were used for SDS polyacrylamide gel electrophoresis and transferred to a nitrocellulose membrane (GE Healthcare) using the Trans-Blot Turbo Transfer System (Bio-Rad, Hercules, CA, USA). Membranes were probed with the following primary antibodies: SIRT1 (H-300) (1:1000) (120 kDa) Santa Cruz sc-15404 and GAPDH (1:2500; Abcam ab9485, Cambridge, UK). After incubation with the corresponding suited horseradish peroxidase-conjugated secondary antibody (1:2000; Abcam ab205718), signals were developed using the enhanced chemiluminescence kit (ECL™ Prime Western Blotting System, GE Healthcare), acquired by ChemiDoc Imaging System XRS+ (Bio-Rad), and analysed for densitometry with the ImageJ software (version 2.0, National Institutes of Health, Bethesda, MD, USA).

### 4.9. Measurements of Cellular Response to TNF-α Challenge 

Two days before bioluminescence measurements, 1–5 × 10^5^ HIF and HCT116 cells, previously transduced with BLH, were seeded in 35 mm dishes. The synchronization was carried out by medium change: cells were washed with 1× PBS (Life technologies) and 2 mL of phenol red-free DMEM (Gibco) containing 10% FBS (Life Technologies), 1% penicillin–streptomycin (Life Technologies), and 250 µM D-Luciferin (PJK) was added. Cells were stimulated by the addition of increasing concentrations of recombinant human TNF-α (Biolegend, San Diego, CA, USA), 25, 50, and 100 ng/mL, to the medium.

### 4.10. Glycolytic Function 

Glycolytic activity was determined using a Seahorse XFe96 Analyzer (Agilent, Santa Clara, CA, USA) and the Seahorse XF Glycolysis Stress Test Kit (Agilent). Cells were seeded one day prior to the assay in 96-well seahorse plates. The assay was performed according to the manufacturer’s instructions. For the glycolysis stress test, seahorse XF base medium (Agilent) supplemented with 2 mM L-glutamine (Thermo Fisher scientific, Waltham, MA, USA) was used, and 10 mM glucose (Agilent), 1 µM oligomycin (Agilent) and 50 mM 2-DG (Agilent) were used for the injections. Subsequently, plates were normalised based on the DNA content using the CyQuant kit (Thermo Fisher scientific) according to the manufacturer’s instructions. Data were analysed using the wave software (Agilent). 

### 4.11. Mitochondrial Function 

Mitochondrial respiration was determined using a Seahorse XFe96 Analyzer (Agilent) and the Seahorse XF Cell Mito Stress Test Kit (Agilent). Cells were seeded one day prior to the assay in 96-well seahorse plates. The assay was performed according to the manufacturer’s instructions. For the cell mito stress test, seahorse XF base medium (Agilent) supplemented with 2 mM L-glutamine (Thermo Fisher scientific), 5.5 mM glucose (Agilent), and 1 mM sodium pyruvate was used, and 2 µM oligomycin (Agilent), 0.5 µM FCCP (Agilent), and 0.5 µM rotenone/antimycin A (Agilent) were used for the injections. Subsequently, plates were normalised based on the DNA content using the CyQuant kit (Thermo Fisher scientific) according to the manufacturer’s instructions. Data were analysed using the wave software (Agilent, Santa Clara, CA, USA). 

### 4.12. Measurement of Cell Viability, Cytotoxicity, and Apoptosis

To determine cell viability, cytotoxicity, and apoptosis, the ApoTox-Glo Triplex Assay (Promega, Madison, WI, USA) was used. Cells were seeded in black 96-well half-area plates with clear bottoms. The assay was performed according to the manufacturer’s instructions. Subsequently, plates were normalised based on the DNA content using the CyQuant kit (Thermo Fisher scientific) according to the manufacturer’s instructions.

### 4.13. Statistical Analysis

Statistical analysis was conducted with the programming language and software environment for statistical computing and graphics in R. Rhythmic variation was tested by fitting the model
$$y=m+a\times \sin\left(2\times \pi \times \frac{t}{\omega }\right)+b\times \cos\left(2\times \pi \times \frac{t}{\omega }\right)$$
to time-series gene and protein expression values using the R package HarmonicRegression [49]. Period ω for the different conditions was chosen dependent on the period lengths observed in the long-term, real-time bioluminescence recording of *BMAL1* promoter activity (HCT116: 20 h, HIF: 25 h, and HCT116+HIF: 22 h).

Co-culture experiments were carried out with at least three biological replicates for each condition. All results are represented as mean ± SEM. Statistical analysis of the results was performed using one-way ANOVA followed by Tukey’s multiple comparisons test. A *p*-value < 0.05 was considered statistically significant (* *p* < 0.05; ** *p* < 0.01; and *** *p* < 0.001).

## 5. Conclusions

Taken together, the results from this study suggest that the interplay of colon cancer cells with tumour-associated stromal cells, in particular TAFs, impacts the molecular clockwork with subsequent alteration of metabolic parameters, improvement of colon cancer cell viability, and decrease of apoptotic response to chemical agents. This can ultimately enhance the malignant phenotype of colon cancer cells and represents a potential target for therapeutic intervention.

## Figures and Tables

**Figure 1 cancers-11-00988-f001:**
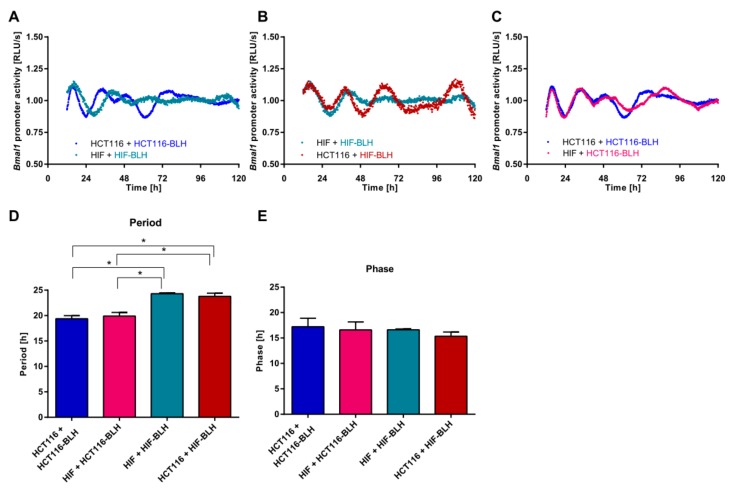
Effect of co-culture on circadian rhythms in HCT116 and human intestinal fibroblast (HIF) cells. Cells were lentivirally transduced and the human *BMAL1*-promoter (BLH) activity was measured over five consecutive days. Cells were either co-cultured with the same cell line (**A**) or with a different cell line (**B**,**C**). Shown is one representative replicate per condition. The BLH-cell line was measured. Period (**D**) and phase (**E**) were calculated in samples with and without co-culture. Data are expressed as mean ± SEM, *n* = 3. Significant changes (*p* < 0.05) between different conditions are marked with *.

**Figure 2 cancers-11-00988-f002:**
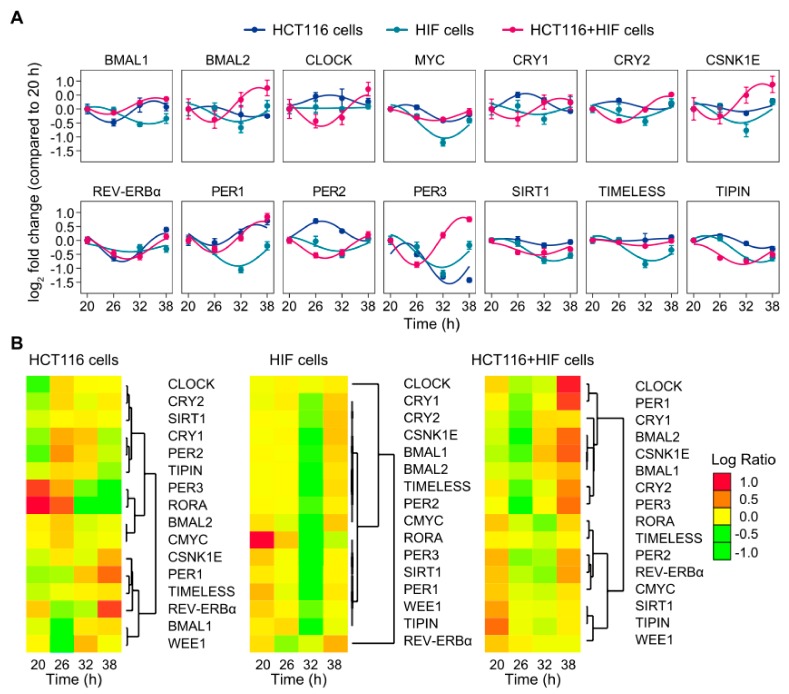
Co-culture of HCT116 and HIF cell lines alters the rhythmic expression of core-clock genes. (**A**) Time-series expression profiles of core-clock genes and putative circadian-regulated genes in HCT116 cells (dark blue), HIF cells (light blue), and a HCT116+HIF co-culture (pink). A sine–cosine curve was fitted to the data using the model $y=m+a\times \sin\left(2\times \pi \times \frac{t}{\omega }\right)+b\times \cos\left(2\times \pi \times \frac{t}{\omega }\right)$. Period ω for the different conditions was chosen dependent on the period lengths observed in the long-term, real-time bioluminescence recording of *BMAL1* promoter activity. (**B**) Hierarchical clustering analysis of sequential transcriptional changes of core-clock genes and clock-controlled genes. Colour code of heatmaps is indicated.

**Figure 3 cancers-11-00988-f003:**
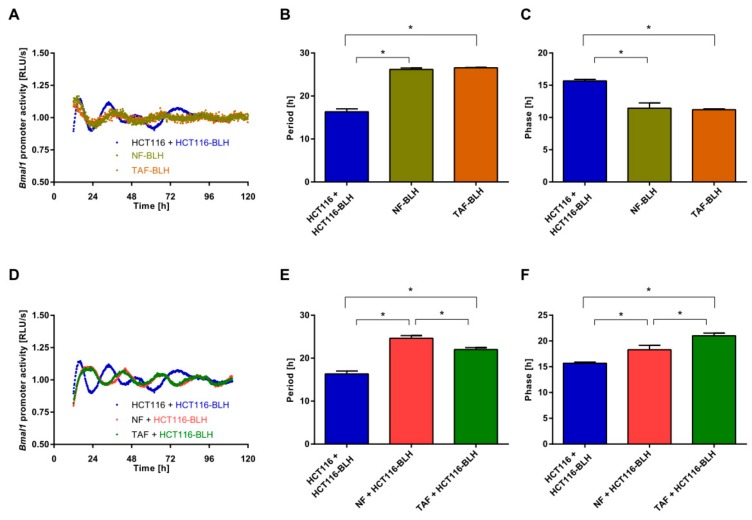
Cell-to-cell communication impacts the circadian phenotype. (**A**) HCT116, normal fibroblast (NF), and tumour-associated fibroblast (TAF) cells were lentivirally transduced and the *BMAL1*-promoter activity was measured over five consecutive days. Shown is one representative replicate per condition. Period (**B**) and phase (**C**) were calculated. Data are expressed as mean ± SEM, *n* = 3. Significant changes (*p* < 0.05) between different cells are marked with *. (**D**) HCT116, NF, and TAF cells were lentivirally transduced and the *BMAL1*-promoter activity was measured over five consecutive days. HCT116 cells were either co-cultured with HCT116 cells or with NFs or TAFs. Shown is one representative replicate per condition. The BLH-cells were measured. Period (**E**) and phase (**F**) were calculated. Data are expressed as mean ± SEM, *n* = 3. Significant changes (*p* < 0.05) between different conditions are marked with *.

**Figure 4 cancers-11-00988-f004:**
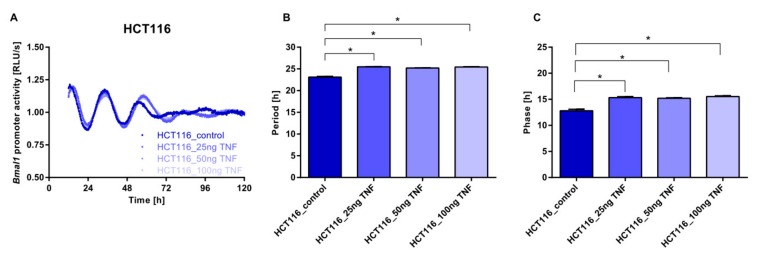
Effect of the stimulation with increasing amounts of human recombinant tumour necrosis factor (TNF) on circadian rhythms in HCT116 cells. HCT116 lentivirally transduced cells were cultured with increasing concentrations of human recombinant TNF (25, 50, and 100 ng/mL), and the *BMAL1*-promoter activity was measured over five consecutive days (**A**). Shown is one representative replicate per condition. Colour gradients represent the different concentrations of recombinant human TNF used. Period (**B**) and phase (**C**) were calculated in the samples and comparisons were made to the control condition. Data are expressed as mean ± SEM, *n* = 3. Significant changes (*p* < 0.05) between different conditions are marked with *.

**Figure 5 cancers-11-00988-f005:**
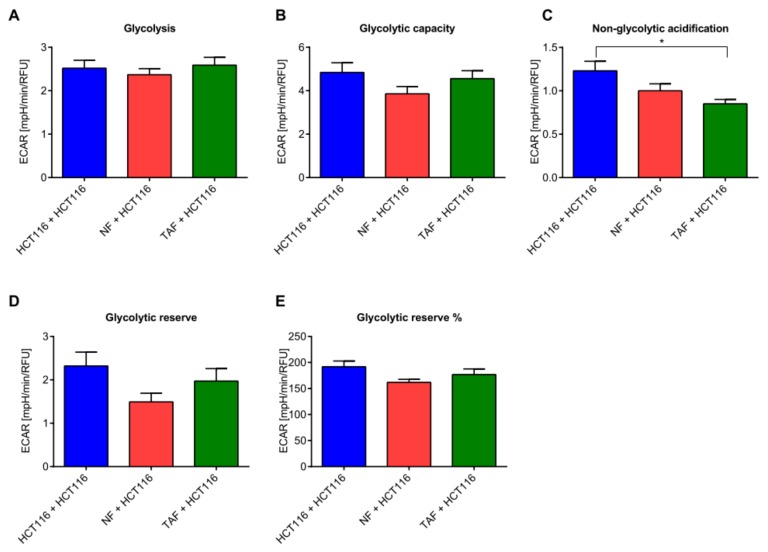
Cell-to-cell communication impacts cell metabolism: glycolysis. (**A**) Glycolysis, (**B**) glycolytic capacity, (**C**) nonglycolytic acidification, (**D**) glycolytic reserve, and (**E**) glycolytic reserve (%) in HCT116 cells cultivated alone or after co-culture with NFs or TAFs. Only HCT116 cells were measured. Data are represented as mean ± SEM (*n* = 8). Significant changes (*p* < 0.05) between different conditions are marked with *.

**Figure 6 cancers-11-00988-f006:**
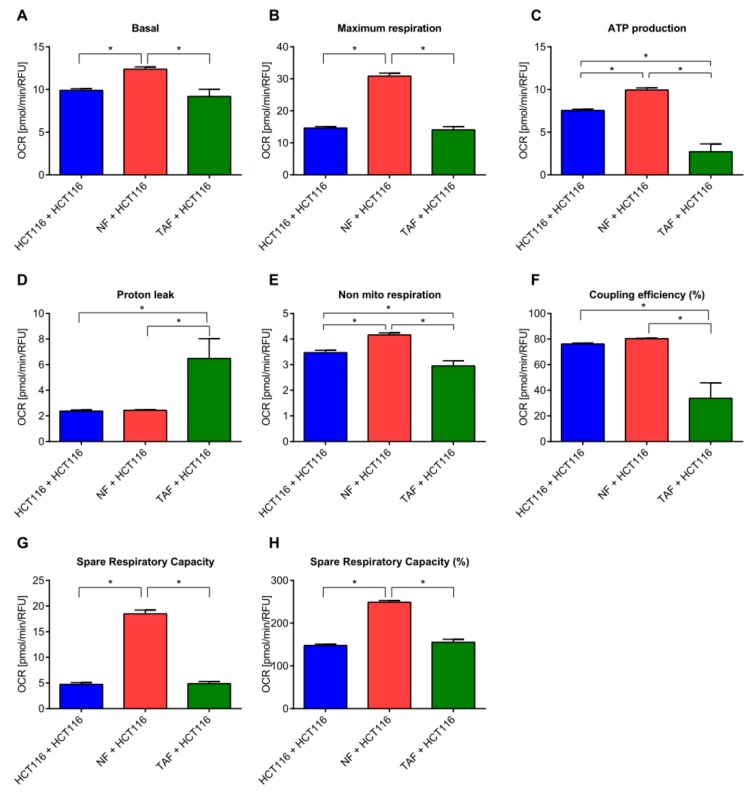
Cell-to-cell communication impacts cell metabolism: mitochondrion. (**A**) Basal respiration, (**B**) maximum respiration, (**C**) ATP production, (**D**) proton leak, (**E**) non-mitochondrial respiration, (**F**) coupling efficiency (%), (**G**) spare respiratory capacity, and (**H**) spare respiratory capacity (%) in HCT116 cells cultivated alone or after co-culture with NF or TAF. HCT116 cells were measured. Data are represented as mean ± SEM (*n* = 8). Significant changes (*p* < 0.05) between different conditions are marked with *.

**Figure 7 cancers-11-00988-f007:**
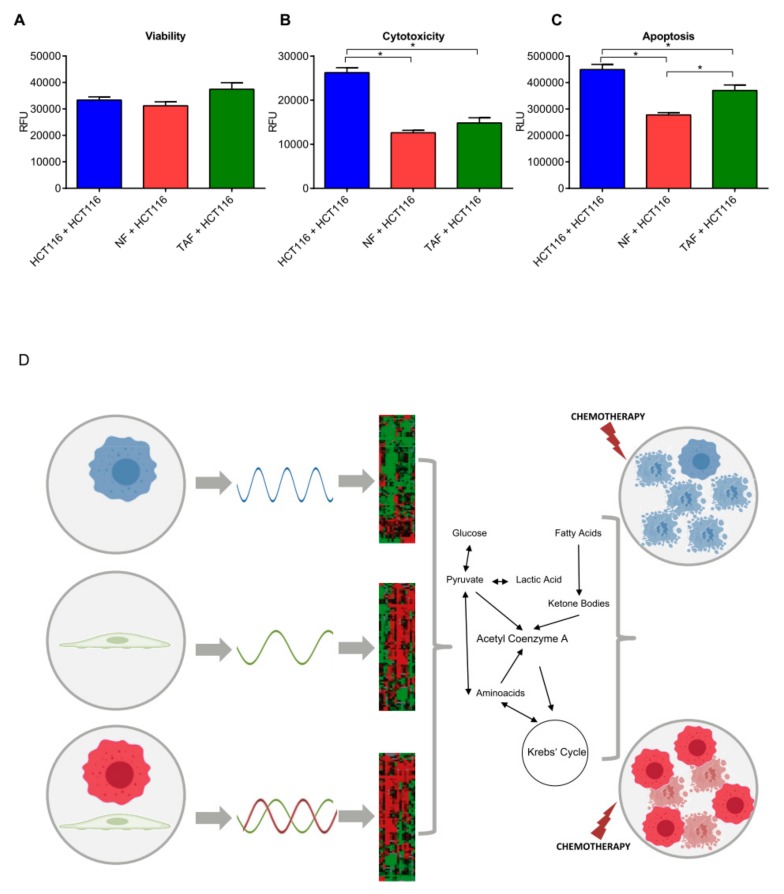
Cell-to-cell communication impacts cancer cell phenotype. (**A**) Viability, (**B**) cytotoxicity, and (**C**) apoptosis in HCT116 cells cultivated alone or after co-culture with NF or TAF. Only HCT116 cells were measured. Data are represented as mean ± SEM (*n* = 5). Statistically significant values (*p* < 0.05) are indicated with *. (**D**) The interaction of cancer cells (top left) with stromal cells (middle left) impacts their circadian phenotype, determines transcriptome modifications, and governs metabolic re-wiring to influence the response to chemotherapy.

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
