# Peer review of "The Interplay between Colon Cancer Cells and Tumour-Associated Stromal Cells Impacts the Biological Clock and Enhances Malignant Phenotypes"

_cancers, 2019, doi:10.3390/cancers11070988_

Round 1

Reviewer 1 Report

This study was well deigned and the results were solid. Once concern could be addressed: the changes of some key circadian genes caused by the co-culture of HCT116 and TAF should be determined or validated in colorectal adenomas and adenocarcinomas.

Author Response

This study was well designed and the results were solid.

We thank this reviewer for the positive evaluation of our work and his/her interesting comments. In the following, we provide a detailed reply to the point raised by the reviewer.

R1.1
Once concern could be addressed: the changes of some key circadian genes caused by the co-culture of HCT116 and TAF should be determined or validated in colorectal adenomas and adenocarcinomas.

Colorectal cancer cell lines display a variety of circadian phenotypes which range from a completely disrupted clock to less dramatic phenotypes, this has been shown in published work [4]. To make this point clear we now include a new supplementary figure (NEW Supplementary Figure 1) where we show the clock phenotype of six different colorectal cancer cell lines and all have a less robust clock as compared to HCT116 cells. For this study we aimed at using a cell line with a robust oscillator (well-defined circadian phenotype) to evaluate how such an oscillator may be affected by the microenvironment, namely by other cells, including primary tumour cells from patients. Thus the HCT116 cell line represents such a robust oscillator and was therefore used in this study.
Additionally, we carried out a new co-culture experiment of the HCT116 cell line with primary cells of a second patient (moderately differentiated colon adenocarcinoma) and observed that also in this new scenario the cells from the cancer patient alter the clock of the HCT116 cells. In particular, there is a significant phase shift of 4.7 hours as a result of the co-culture with NF and of 6.2 hours upon co-culture with TAF. In addition, the period of the oscillation shows alterations although not significant (New Supplementary Figure 4). We include both figures in the manuscript.

Changes to the manuscript:

Results
Colorectal cancer cell lines display a variety of circadian phenotypes which range from a completely disrupted clock to less dramatic phenotypes, this has been shown in published work [4]. For this study we aimed at using a cell line with a robust oscillator (well-defined circadian phenotype) to evaluate how such an oscillator may be affected by the microenvironment, including primary tumour cells from patients. The HCT116 cell line represents a robust oscillator as compared to other colorectal cancer cell lines was therefore used in this study (Supplementary Figure 1).
Additionally, we carried out a new co-culture of the HCT116 cell line with primary cell of a second patient (with moderately differentiated colon adenocarcinoma) and observed that also in this new scenario the cells from the cancer patient alter the clock phenotype of the HCT116. In particular, there is a significant phase shift of 4.7 hours as a result of the co-culture with NF and of 6.2 hours upon co-culture with TAF. Also the period of the oscillation shows alterations although not significant (Supplementary Figure 4).

Reviewer 2 Report

The manuscript Manuscript ID: cancers-536120 titled “The interplay between colon cancer cells and tumor-associated stromal cells impacts on the biological clock and enhances malignant phenotype” by Luise Fuhr et al have explored the novel role of the biological clock on the colon cancer malignancy. This study has been well documented with supportive data. In this study, the author used colon cancer cells (HCT-116), commercially available normal human intestinal fibroblasts (HIF) as well as primary normal fibroblasts (NFs), and primary tumor-associated fibroblasts (TAFs) derived from CRC patients to test their hypothesis. The author implied the techniques such as the production of lentivirus and transduced the cells with lentiviral vectors, measurement of cell bioluminescence, qPCR for mRNA level, measured the cellular response to TNF challenge, glycolytic function, mitochondria function, cell viability, cytotoxicity, and apoptosis using the specific commercially available kit in this manuscript. The minor comments are described below:

1.    This study would be strengthened if the author tested the preliminary findings using another colon cancer cell line as the cell to cell communication impacts on the circadian phenotype.

2.    It is not clear if this interplay exists in the stem cells and tumor-associated stromal cells as the stem cells play a vital role for malignancy.

3.    This study is limited to the transcription level. It is suggested to verify the transcript level with the translational level with at least 2 or 3 proteins by western blot.

4.    It is unclear to the reviewer the number of patients involved in this study.

5.    In the methodology section, it is mentioned that normal colon fibroblasts and tubulous villous adenoma-associated fibroblasts (TAF) were derived from the same patient. But it is not mentioned which cancer stage is being involved. Does this study have any impact on the colon cancer stage (stage I-IV) progression?

6.    In the text, page 3 second paragraph, the author evaluated 18 h time-course measurements of mRNA expression levels which do not correspond with the Figure 2A and B. In this figure, the author demonstrated the alteration of core-clock genes with different time point (20-38h).

Author Response

Please see the responses to Reviewer #2 in the attachment. Thanks.

Reviewer 3 Report

In this research paper the authors have studied the effect of co-culturing cancer cells with tumor associated fibroblasts isolated from tubulovillus adenoma.  They showed that co-culturing with tumor associated fibroblasts decreased cancer cell apoptosis, increased viability and also increased resistance to chemotherapeutic agents.  This study clealy shows that tumor associated fibroblasts influence cancer cell malignancy and the phenotype.

Comments:

  The authors can discuss some potential mechanisms by which cancer associated fibroblasts influence the cancer cells (HCT-116) in the discussion.

It would be helpful to also do the experiments using few other cancer cell lines.

Author Response

Please see the responses to Reviewer #3 in the attachment. Thanks.

Round 2

Reviewer 3 Report

The revised manuscript is satisfactory and addresses the concerns raised by the reviewers.

Author Response

We are happy to see that the reviewer is satisfied with the revised version of our work and thank the reviewer for her/his comments.